# HDAC Inhibitors Induce *BDNF* Expression and Promote Neurite Outgrowth in Human Neural Progenitor Cells-Derived Neurons

**DOI:** 10.3390/ijms20051109

**Published:** 2019-03-05

**Authors:** Amir Bagheri, Parham Habibzadeh, Seyedeh Fatemeh Razavipour, Claude-Henry Volmar, Nancy T. Chee, Shaun P. Brothers, Claes Wahlestedt, Seyed Javad Mowla, Mohammad Ali Faghihi

**Affiliations:** 1Department of Molecular Genetics, Faculty of Biological Sciences, Tarbiat Modares University, Tehran, P.O. Box 14115-111, Iran; axb1907@med.miami.edu; 2Center for Therapeutic Innovation and Department of Psychiatry & Behavioral Sciences, University of Miami Miller School of Medicine, Miami, FL 33136, USA; cvolmar@miami.edu (C.-H.V.); ntm30@med.miami.edu (N.T.C.); sbrothers@umail.miami.edu (S.P.B.); CWahlestedt@med.miami.edu (C.W.); 3Persian BayanGene Research and Training Center, Shiraz, P.O. Box 7134767617, Iran; Parham.Habibzadeh@yahoo.com; 4Department of Biochemistry and Molecular Biology, University of Miami Miller School of Medicine, Miami, FL 33136, USA; sxr986@med.miami.edu

**Keywords:** HDAC inhibitor, BDNF, neurite outgrowth, human neural progenitor cell, neurodegenerative disorder, epigenetic library, screening

## Abstract

Besides its key role in neural development, brain-derived neurotrophic factor (BDNF) is important for long-term potentiation and neurogenesis, which makes it a critical factor in learning and memory. Due to the important role of BDNF in synaptic function and plasticity, an in-house epigenetic library was screened against human neural progenitor cells (HNPCs) and WS1 human skin fibroblast cells using Cell-to-Ct assay kit to identify the small compounds capable of modulating the *BDNF* expression. In addition to two well-known hydroxamic acid-based histone deacetylase inhibitors (hb-HDACis), SAHA and TSA, several structurally similar HDAC inhibitors including SB-939, PCI-24781 and JNJ-26481585 with even higher impact on *BDNF* expression, were discovered in this study. Furthermore, by using well-developed immunohistochemistry assays, the selected compounds were also proved to have neurogenic potential improving the neurite outgrowth in HNPCs-derived neurons. In conclusion, we proved the neurogenic potential of several hb-HDACis, alongside their ability to enhance *BDNF* expression, which by modulating the neurogenesis and/or compensating for neuronal loss, could be propitious for treatment of neurological disorders.

## 1. Introduction

Alzheimer’s disease (AD), Parkinson’s disease (PD), and amyotrophic lateral sclerosis (ALS) are the major neurodegenerative diseases imposing a considerable burden on health care systems worldwide. With improved life expectancy in recent years in many countries, the prevalence and incidence of these age-related diseases is also on the rise [1].

Reduction in histone acetylation is considered to be a common cause of various brain pathologies such as neurodegenerative and neurodevelopmental cognitive disorders [2]. In the wake of studies showing the key role of histone acetylation in memory formation, researchers have investigated the HDACis potential to improve memory formation in diseases characterized by memory loss and cognitive impairments such as AD. It is now well documented that various HDACis are able to improve the memory and cognition in mice and rats [2].

HDACis may be beneficial in the treatment of various brain disorders through three potential mechanisms: (1) anti-inflammatory effect via down-regulation of pro-inflammatory factors such as Fas-L and IL-6; (2) anti-neurotoxic effects through down-regulation of neurotoxic proteins such as α-synuclein and β-amyloid; and (3) neuroprotective effect via up-regulation of neuroprotective proteins such as brain-derived neurotrophic factor (BDNF) and the glial cell-derived neurotrophic factor (GDNF) [3,4,5].

BDNF is a secreted protein belonging to the neurotrophin family of growth factors that binds to two distinct receptors, p75 and TrkB, differing in their respective downstream signaling pathways [6,7]. Besides playing a key role in neural development, BDNF is important for long-term potentiation and neurogenesis, which makes it critically important in learning process and memory formation as well as reward-related processes [8,9,10,11]. Due to its effect on neuronal and synaptic plasticity, BDNF also has a significant role in the pathogenesis of psychiatric disorders [12]. Compromised *BDNF* expression regulation has been found to be associated with cognitive and neurodegenerative diseases, early-life adverse experience, mood and anxiety disorders, and aging [13,14,15,16,17,18].

To investigate the role of BDNF in synaptic function and plasticity, it was selected as the target gene for our screening project against an in-house epigenetic library for the following reasons: Firstly, unlike neuronal loss, synaptic dysfunction and loss are reversible; Secondly, regardless of the type and cause of the damaging insult to the synaptic structure and function as the causative factor for the development of various psychiatric and neurological disorders, it may be a promising site for therapeutic interventions; Lastly, a great challenge regarding the treatment of neurodegenerative disorders is the delayed diagnosis, which makes the available treatments less effective. However, the therapeutic interventions acting through synaptic repair have a wider window for action and can thus be used at later stages of the disease, which makes them a great hope for novel and effective therapeutic modalities [19,20,21].

Neurite outgrowth is a crucial process during neurogenesis. Neurite outgrowth has an important role in neurodevelopment and is considered to be responsible for neuronal connectivity during brain development, disruption of which could result in cognitive deficits [22,23,24,25]. The capacity of neurons to induce and extend neurite projections can be used as a parameter to determine the specific chemical compounds ability to produce neurogenesis.

This study was therefore conducted to identify compounds capable of modulating the expression of *BDNF* and subsequently evaluate their effects on neurite outgrowth as the process in which BDNF is known to play a key role. In our work, an in-house epigenetic library was screened against human neural progenitor cells (HNPCs) and WS1 human skin fibroblast cells using Cell-to-Ct assay kit to identify the factors modulating the *BDNF* gene expression. The compound library used consisted of well-defined histone deacetylase (HDAC) inhibitors, lysine demethylase (KDM), histone acetyltransferase (HATs), DNA methyltransferase (DNMTs), and epigenetic reader domain inhibitors (e.g., BET inhibitors).

## 2. Results

### 2.1. Primary Screenings for Enhancers of BDNF mRNA Expression

HNPCs and WS1 human skin fibroblast cells were treated in 384-well plates and screened against an in-house epigenetic library with more than 160 compounds at 1 µM concentration (Figure 1).

To maximize the available amount of RNA, Cell-to-Ct assay kit was used to extract the RNA and to make corresponding complementary DNA (cDNA). We found that 67 and 57 compounds were effective on the *BDNF* expression in HNPCs and fibroblast cells, respectively. Z-factor, as a measure of assay suitability for high-throughput screening, was determined. A Z-factor > 0.5 for primary screening in HNPCs was considered a highly reliable assay. The coefficient of variation was another standard parameter used to measure the suitability of the assay for high-throughput screening (HTS); a coefficient of variation <10% was used as a primary screening for HNPCs experiment suitability for HTS.

### 2.2. HNPCs and Fibroblasts Shared Common Hits

HNPCs and fibroblast cells shared several common small molecules belonging to the same functionally related category (Table 1 and Figure 2). Most (19 of 26) of the shared compounds belonged to HDACis category. Methyltransferase inhibitors and PARP-1 inhibitors, each with two compounds, were two other functionally related categories. HAT inhibitor, histone demethylase inhibitor and natural compound, sharing only one compound in each category, are also contributing in this list.

### 2.3. HDACis Increased BDNF in Secondary Assays

To confirm the observed data from primary screenings, the effective compounds were tested in Hek293 to assess their ability to increase the *BDNF* gene mRNA expression (Figure 3). Trichostatin A was used as positive control for its validated effect on overexpression of *BDNF* [26,27]. JNJ-26481585, PCI-24781 and SB-939 were used as putative *BDNF* expression enhancers shared between HNPCs and fibroblast cells. Ellagic acid, Droxinostat and Entacapone were chosen as neutral controls, and were proved not to be effective in primary screening tests. Moreover, we used another compound, I-Bet 151, which was shown to have the opposite effects on *BDNF* expression in HNPCs and fibroblast cells—increasing *BDNF* mRNA expression in HNPCs but decreasing it in fibroblasts. qRT-PCR experiments with TaqMan probes confirmed test compounds to be effective in a dose-dependent manner, enhancing the *BDNF* mRNA expression with elevated concentrations. In case of I-Bet 151, the *BDNF* mRNA expression was decreased in Hek293 cells, not confirming the effect observed in HNPCs but replicating the effect observed in fibroblast cells (data not shown). Since the mRNA level in a cell is not always proportional to the amount of the corresponding protein, we decided to perform an ELISA test to measure BDNF protein level upon treating the Hek293 cell line with test compounds that was proven to be effective on *BDNF* mRNA expression (Figure 4). All selected compounds significantly elevated the BDNF protein level, most notably for SB-939 and JNJ-26481585 with around 30 pg/mL for both compounds at 1 and 0.1 µM concentrations, respectively. Finally, a set of three compounds, all belonging to hydroxamic acid-based HDACis, were verified from the combination of primary screening and secondary validations for further experiments.

### 2.4. Neurite Length Was Significantly Enhanced upon Treatment with Hydroxamate-Based HDACis

We showed that test compounds could increase the *BDNF* expression at mRNA and protein levels. We subsequently investigated whether the test compounds would be effective on neurite outgrowth in HNPCs-derived neurons (Figure 5). Since TSA has been reported to increase the neurite length, it was first checked and upon confirmation, used as a positive control [28]. Neurons were labeled with anti-β-tubulin III antibody and the lengths of neurites were measured 24 h after the treatment. With an average neurite length of 73.5 µm, the cells treated with PCI-24781 at a concentration of 0.1 µM had the longest neurons—37% more than the cells treated with 0.1% DMSO (as control). JNJ-26481585 at a concentration of 0.001 µM resulted in the second highest increase in the length of neurites by 72.3 µm—34% more than that with the control. Despite slight differences observed, all test compounds were demonstrated to be significantly effective on the neurite length.

### 2.5. The Hydroxamate-Based HDACis Showed No Toxicity at the Concentrations Used

Except for JNJ-26481585, which was tested at a concentration of 0.001 µM, other compounds were tested at two different doses—0.01 and 0.1 µM—for cell toxicity. Neuronal cells viability was assessed with Cell-Titer Glo assay. HNPCs were seeded and differentiated to neurons according to the same exact method used for neurite outgrowth assay. The viability did not significantly change relative to the control (0.1% DMSO) in all test compounds (Figure 6).

## 3. Discussion

Due to the improved physiological relevance and translational potential, the use of primary human cells as experimental cell model offers significant value in drug discovery assays. While these cells provide an ideal system for screening of compounds, their maintenance and scaling for HTS is challenging [29,30,31,32]. Herein, we provided evidence showing that HNPCs could successfully be adapted and scaled to HTS in 384-well plates. HNPCs are comprised of neurospheres, floating cellular aggregates containing neural progenitor cells (NPCs) and radial glial cells (RGCs) isolated from embryonic mammalian brain that have been cultured in vitro in the presence of mitogens. These cells maintain their differentiation capability in creating different neural lineages providing a desirable in vitro model system to study the CNS function and development [33,34,35]. In addition to the differentiation capability of neurospheres, their undifferentiated state is also previously confirmed by detecting the expression of the typical progenitor cell markers containing BLBP, NESTIN, VIM, and GFAP [26].

In this experiment we showed that several hb-HDACi compounds including SB-939, PCI-24781 and JNJ-26481585 were powerful inducers of *BDNF* expression in mRNA and protein level and were also able to induce the neurite outgrowth in human neural progenitor cells-derived neurons, demonstrating their neurogenic potential.

In addition to HDAC inhibitors, several other distinct molecular categories were also shown to be potent activators of *BDNF* gene expression in both HNPCs and fibroblasts in our primary screening experiments (Table 1). MM-102 and 5-Aza-2′-deoxycytidine (Decitabine) are methyltransferase inhibitors acting on histone and DNA, respectively. It is demonstrated that *BDNF* down-regulation in brain samples taken from patients with schizophrenia and bipolar affective disorder is associated with DNA methyltransferase1 (DNMT1) overexpression [36]. Therefore, using 5-Aza-2′-deoxycytidine (Decitabine) as a DNMT1 inhibitor could be a promising strategy to tackle neurodegenerative disorders. MM-102 is also a small molecule inhibitor of WDR5/MLL1 protein-protein interaction that specifically inhibits cell growth and induces apoptosis in leukemia cells harboring MLL1 fusion proteins [37]. PARP-1 inhibition has been reported to play a neuroprotective role in neurodegeneration induced by acute brain ischemia [38]. Its association with the pathogenesis of several nervous system disorders has also been demonstrated [39,40,41]. Due to the inhibitory effects of nicotinamide and BYK 204165 on PARP-1, these compounds may thus be able to demonstrate promising effects on the models of neurodegenerative disorders. CPTH2 inhibits the HAT activity of Gcn5. Histone acetyl transferase function usually leads to transcriptional activation, but in this study it showed an activation effect on the expression of *BDNF*, which once more draws attention to the complicated regulatory mechanisms of BDNF [42]. 2,4-pyridinedicarboxylic acid (2,4-PDCA), as a histone demethylase inhibitor, has been shown to be able to inhibit several Jumonji domain-containing lysine demethylases when used at low micromolar concentrations [43,44,45]. It should be noted that there is no documented information about the effect of this compound on *BDNF*. Therefore, this compound can be considered as a putative *BDNF* enhancer to investigate its positive effects in neurodegenerative disorders. Resveratrol is a natural nonflavonoid polyphenol found in grapes and red wine. It has anti-neuroinflammatory properties and is known to be neuroprotective in neurodegenerative disorders [46,47]. The enhancing effect of resveratrol on *BDNF* gene expression has been validated by in vitro and in vivo experiments elsewhere [48,49,50,51].

*BDNF* has a complex structure, containing nine exons (I–IX) in both human and rodents with each exon harboring its own promoter, resulting in more than 10 different alternatively spliced transcripts [52,53,54,55]. Temporal and spatial regulation of *BDNF* gene expression occurs by using distinct *BDNF* mRNA spliced variants and is also dependent on its various promoters, which could finally lead to the modulation of synaptic plasticity and spine development in dendrites [56,57,58]. *BDNF* gene expression is also controlled at the post-transcriptional level [13].

We demonstrated here that *BDNF* gene was reactive to a diverse array of epigenetic factors, most notably HDAC inhibitors, more specifically hydroxamic-based HDAC inhibitors. We believe that the information obtained from our screening could pave the way for identification of novel treatments for neurodegenerative disorders. In addition, our findings shed light on the complex regulatory mechanisms of *BDNF* gene and reanalysis of the data would likely lead to identification of new pathways that are important to the regulation of this gene.

Neurite outgrowth plays a key role in the regeneration of the nervous system following injury. It is also an integral element of extracellular signaling that can induce neuronal regenerative activities to enhance the outcomes of cases with neurodegenerative disorders and neuronal injury [59,60,61,62]. In this experiment, we investigated the effect of hb-HDACis on the neurite outgrowth. The rationale behind this analysis was that since BDNF is an important factor capable of inducing neurite outgrowth, the compounds which we proved to be *BDNF* activators may likewise affect the neurite outgrowth. In this investigation, hb-HDACis was demonstrated to promote neurite outgrowth even more than that induced by TSA, a documented positive control for this specific experiment. Neurite outgrowth might be the direct result of BDNF production; however, this hypothesis cannot be proved unless TrkB signaling pathway is separately blocked when the identified HDAC inhibitors are applied.

## 4. Materials and Methods

### 4.1. High-Throughput Screening Assay Using an In-House Epigenetics Compound Library

HNPCs and WS1 human skin fibroblast cells (ATCC CRL-1502) were seeded in 384-well plates (25K and 20K cells per well, respectively) and treated with an in-house epigenetic compound library at a concentration of 1 µM in duplicate. To maximize the amount of available RNA, TaqMan Cell-to-Ct assay kit (Invitrogen AM1729) was used to extract the RNA and reverse transcription to make the resulting cDNA per the manufacturer’s instruction. In short, after aspiration of the media, lysis solution (30 µL) with diluted DNase I was added; then the plate was shaken for 2 min and incubated for 3 min at room temperature. Subsequently Stop solution (3 µL) was added and the solution was incubated at room temperature for 2 min. After spinning, 4 µL of lysates was transferred to a new PCR plate with 16 µL of reverse transcription enzyme mix previously added to each well. The thermal cycling condition was as follows: 60 min at 37 °C, and 5 min at 95 °C. A 3-µL aliquot of each cDNA reaction was then added to 13 µL of each TaqMan master mix reaction along with TaqMan *BDNF* probe (Hs02718934_s1 Invitrogen 444889). A QuantStudio 6 Flex Real-Time PCR system (Applied Biosystems) was utilized to determine the Ct values. Relative mRNA expression levels were normalized to β-actin (Invitrogen 4326315E) and analyzed using the comparative delta-delta CT method.

### 4.2. Neurite Outgrowth Experiment

#### 4.2.1. Isolation and Culture of Human Neural Progenitor Cells (HNPCs)

HNPCs were isolated from human fetal brain collected from a second trimester-aborted fetus. HNPCs were obtained, isolated and cultured as previously described [33,63]. In short, fetal brain tissue was mechanically dissociated into single cells, seeded in 75-mm tissue culture flasks provided with neurobasal media (Gibco 21103049) supplemented with EGF (20 ng/mL) (Gibco PHG0311), FGF (10 ng/mL) (Gibco PHG6015), B27 (Thermo 12587010), Glutamax (Gibco 35050061) and heparin (2 μg/mL) (Calbiochem 375095). Following 7–10 days in culture, neural stem cells (NSCs) form free-floating neurospheres that could be cultured for several months.

#### 4.2.2. Differentiation and Treatment of HNPCs

To induce differentiation, cultured neurospheres were disaggregated into single cells using StemPro Accutase cell dissociation reagent (Gibco A1110501), counted with Countess Automated Cell Counter (Invitrogen, Carlsbad, CA, USA C10227), then 80K NPCs were plated in each well of 4-well glass chamber slides (Millipore Sigma, Burlington, MA, USA PEZGS0816) previously coated with poly-l-lysine (PLL) (Sigma P5899-5MG) for 1 hour at room temperature and laminin (Sigma L2020-1MG) for 2 h at 37 °C for 5 days. The differentiation media consisted of DMEM/F12 (Gibco, Waltham, MA, USA 11320033) supplemented with N2 (1%) (Gibco 17502048), MEM non-essential amino acids (1%) (Gibco 11140050) and heparin (2 µg/µL) (Calbiochem 375095) and containing B-27 (Thermo, Waltham, MA, USA 12587010), antibiotic-antimycotic (Gibco 15240062), retinoic acid (1 mM) (Sigma R2625), GDNF (10 µg/µL) (PeproTech, Rocky Hill, NJ, USA 450-10), BDNF (10 µg/µL) (PeproTech 450-02) and l-ascorbic acid (10 µg/µL) (Sigma A92902-25G). After 5 days in culture, differentiated cells were then treated with the following compounds: DMSO (Sigma 34869), PCI-24781 (Cayman Chemical, Ann Arbor, MI, USA 20059), SB-939 (Cayman Chemical 10443), trichostatin A (Cayman Chemical 89730), JNJ-26481585 (Cayman Chemical 14088) for 24 h before immunostaining and fluorescence quantification.

#### 4.2.3. Immunocytochemistry (ICC)

Cells were fixed with 4% formaldehyde (Thermo FB002) for 15 min and were subsequently washed twice with PBS. In order to prevent non-specific binding of primary antibodies, permeabilization and blocking was performed for 1 hour at room temperature using 2% Triton X-100 and 20% goat serum containing buffer consisted of distilled water, NaCl (150 mM) (Sigma 71376), Tris Base (50 mM) (Sigma 10708976001), BSA (1%) (Sigma A8531), l-lysine (100 mM) (SIGMA L5501), Sodium azide (0.04%) (Sigma S2002), pH 7.4. Cells were then incubated with primary antibodies, anti-β-tubulin III (1:200) (Thermo MA1-118X) and Synaptophysin (Thermo MA5-14532), overnight at 4 °C. Subsequently, cells were washed twice with PBS and incubated with fluorescently labeled secondary antibodies, Alexa Fluor 488 (1:500) (Invitrogen A-11001) and Alexa Fluor 594 (2 drops/mL) (Invitrogen R37117), in dark place for 2 h at room temperature. After washing twice with PBS, a Dapi containing mounting media (Thermo S36964) was used for staining the nucleus and mounting the slides.

#### 4.2.4. Image Acquisition and Neurite Outgrowth Quantification

Following staining, the images were captured with a LSM 710 confocal microscope equipped with ZEN 2010 B SP1 software (Carl Zeiss, Oberkochen, Germany), using the 20× objective with an image size of 1024 × 1024 pixels and 424.7 × 424.7 μM. At least two fields per well containing stained differentiated neuronal cells were imaged.

Fiji image analysis software (ImageJ 1.51u) was used for quantification. Briefly, regions of interest (ROIs) and the length of the longest neurite from each neuron were selected and then ‘Measure’ function was used to quantify the length of the selected ROIs. Subsequently, the values per treatment were averaged. Error bars indicate SEM of neurite length calculated from four biological replicates per treatment (at least two images per replicate). Student’s *t* test for independent groups was used to compare means between two groups. A two-tailed *p* value < 0.05 was considered statistically significant.

#### 4.2.5. RNA Extraction, cDNA Synthesis, and Real-Time PCR

Total RNA was extracted from HEK293 cells using the RNeasy Mini Kit (Qiagen, Hilden, Germany 74106) with on-column performed deoxyribonuclease (DNase) treatment using a Ribonuclease-Free DNase Kit (Qiagen), according to the manufacturer’s protocol. The quality of RNAs was checked using NanoDrop 2000 spectrophotometer (Thermo Scientific). RNA (1 µg) was then reverse-transcribed using TaqMan reverse transcription reagents (Invitrogen, Cat No. N8080234). Gene expression was then measured by real-time PCR (RT-PCR) using human *BDNF* (Invitrogen, cat No. 444889, assay ID Hs02718934_s1) and human ACTB (β-actin) endogenous control (Invitrogen, cat No. 4326315E) TaqMan primer-probe assays and TaqMan gene expression master mix (Invitrogen 4369016). Samples were amplified for 40 cycles using the Applied Biosystem Quantstudio Flex 6 Real-Time PCR System (Applied Biosystems, Foster City, CA, USA). The relative gene expression presented was calculated based on fold change using the delta *C*_t_ method. Data were analyzed using two-tailed Student’s *t* test with GraphPad Prism software (San Diego, CA, USA).

#### 4.2.6. Cell Culture Drug Treatment

HEK293 cells, were used to test the effect of the compounds in vitro. Concisely, cells were plated overnight in DMEM supplemented with 10% fetal bovine serum in 24-well plates (200,000 cells per well) in a 37 °C incubator, 5% CO_2_, treated the next day with DMSO (0.1%) or selected compounds (1 µM) for 48 h.

#### 4.2.7. Cell Viability Assay

Cytotoxicity of selected compounds was evaluated using the CellTiter-Glo luminescent cell viability assay (Promega G7570) as per the manufacturer’s protocol. For neuronal cells, the method used was the same as described in “Differentiation and treatment of HNPCs” section. Subsequently, ATP level was measured after treatment utilizing EnVision 2104 Multilabel Reader (Perkin Elmer, Waltham, MA, USA). The luminescent signal is proportional to the cellular ATP concentration which itself is directly proportional to the cell number. Error bars represent SD. One-way ANOVA was used for data analysis. A *p* value < 0.05 was considered statistically significant.

#### 4.2.8. ELISA Experiments

HEK293 cell culture supernatants were collected 48 h after treatment with compounds. Levels of secreted BDNF were measured using BDNF Emax ImmunoAssay System per manufacturers’ instructions (Promega G7611). Samples were measured in triplicate, and the mean value was used for analysis. Error bars represent SEM of BDNF protein. Statistical significance was calculated using two-tailed Student’s *t* test for unpaired data. A *p* value < 0.05 was considered statistically significant.

## Figures and Tables

**Figure 1 ijms-20-01109-f001:**
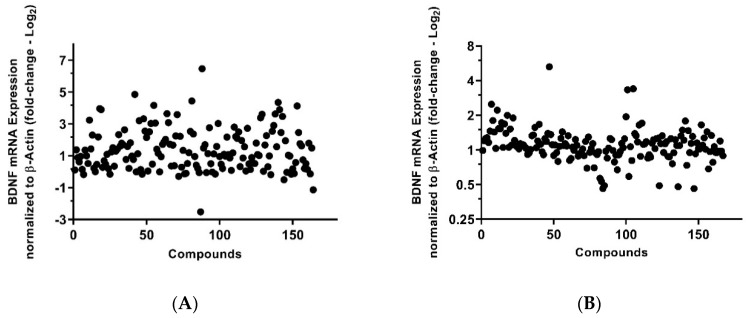
Scatterplot of the effect of Epigenetics Library compounds (1 µM, 0.1% DMSO) on the *BDNF* mRNA expression. (**A**) The primary screen in HNPCs against 164 epigenetic compounds. 67 potent *BDNF* expression effectors (activator and inhibitor) were identified. Z-factor = 0.55, Hit rate = 39.39%, Coefficient of variation = 2.48%; (**B**) The primary screen in human fibroblasts against 167 epigenetic compounds. 57 potent *BDNF* expression effectors (activator and inhibitor) were identified. Z-factor = 0.046, Hit rate = 34.73%, Coefficient of variation = 4.51%. Data points are the average of duplicates expressed as fold change (Log_2_) *BDNF* mRNA expression relative to the control (DMSO 0.1%). The threshold for selection of hits for activators was set at 3 SD from the mean (≥1.4-fold change).

**Figure 2 ijms-20-01109-f002:**
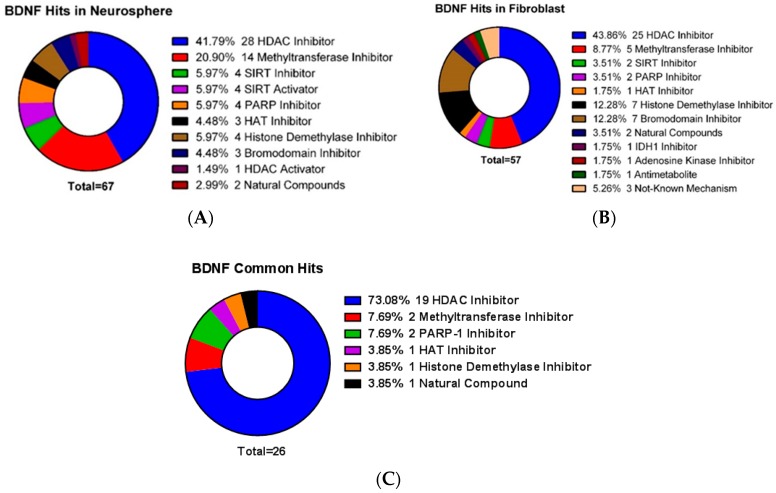
Donut charts demonstrating the *BDNF* effectors categorized according to their molecular function. (**A**) In HNPCs HDAC inhibitors by having 42% (28 of 67 compounds) are the most notable group. Methyltransferase inhibitors are the second most notable category by having 21% (14 of 67 compounds). (**B**) In fibroblasts, the most notable group is HDAC inhibitors, having 44% (25 of 57 compounds). Histone demethylase inhibitors and bromodomain inhibitors are the second most notable categories by having 12% (each with 7 out of 57 compounds). (**C**) Common compounds between HNPCs and fibroblasts affecting *BDNF* mRNA transcription. HDAC inhibitors by having 19 of 26 compounds had the highest rank among various categories studied. The second rank belongs to methyltransferase inhibitors and PARP-1 inhibitors each with 2 compounds.

**Figure 3 ijms-20-01109-f003:**
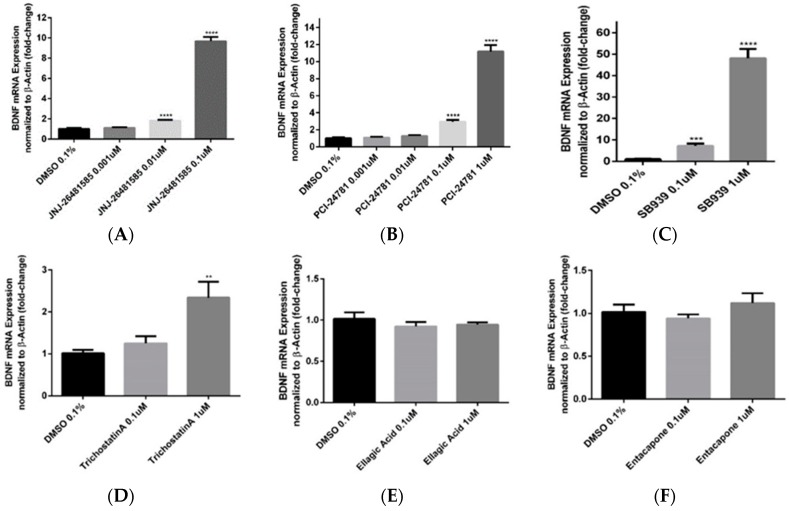
*BDNF* mRNA expression level in HEK293 cells upon treatment with various compounds as determined by RT-qPCR analysis. (**A**–**C**) JNJ-26481585, PCI-24781, and SB939 significantly increased the amount of *BDNF* mRNA expression. (**D**) Trichostatin A moderately increased the *BDNF* mRNA expression. (**E**–**G**) In case of ellagic acid, entacapone, and droxinostat, the level of *BDNF* mRNA was either not changed at all or the change was not significant. (**H**,**I**) Bet 151 was shown to be a negative control for *BDNF* mRNA expression, decreasing its amount significantly. Data were normalized to β-actin. Three biological replicates were used per treatment; Error bars represent SEM; ** *p* < 0.01, *** *p* < 0.001, **** *p* < 0.0001.

**Figure 4 ijms-20-01109-f004:**
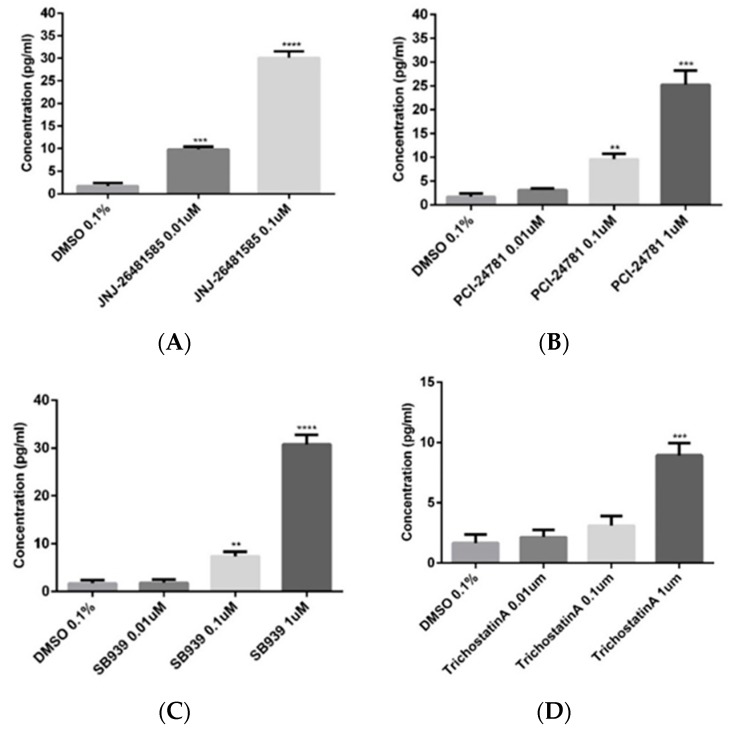
ELISA results showing that BDNF protein was significantly increased in HEK293 cells upon treatment with JNJ-26481585, PCI-24781, and SB939 compounds. Trichostatin A was used as a positive control. Three biological replicates were used per treatment; Error bars represent SEM; ** *p* < 0.01, *** *p* < 0.001, **** *p* < 0.0001.

**Figure 5 ijms-20-01109-f005:**
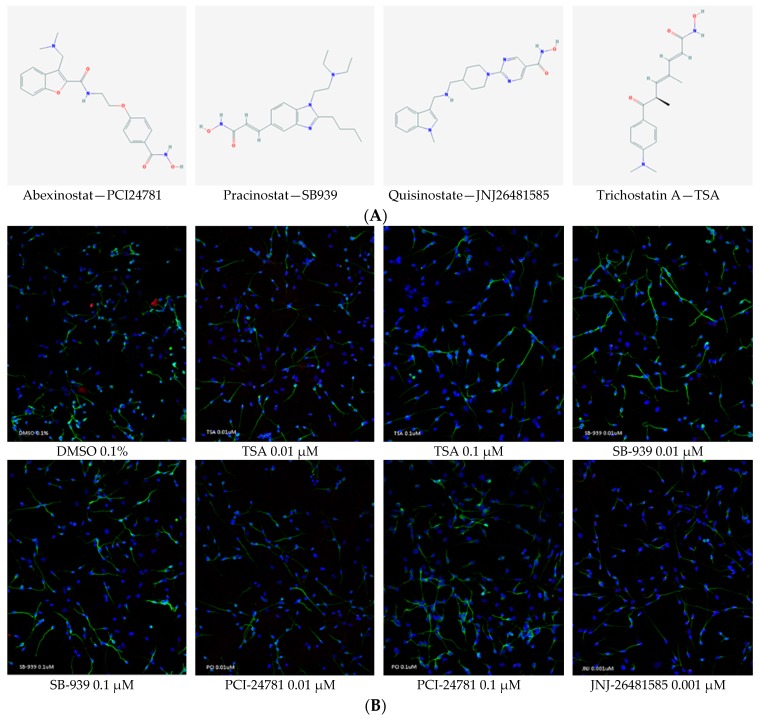
(**A**) Chemical structures of the test compounds; abexinostat (PCI24781), pracinostat (SB939), quisinostat (JNJ26481585), and trichostatin A (TCA). (**B**) Representative fluorescent images (20× magnification) for each treatment. Human neural progenitor cells were differentiated for 5 days; then treated for 24 h with 0.1% DMSO (as control), and trichostatin A, JNJ26481585, SB939, and PCI24781 (as test compounds). Then immunostaining was performed with neuronal-specific β-tubulin III antibody (green) to visualize neuronal processes and quantify neurite lengths and Dapi (blue) to visualize cell bodies. (**C**) Statistical analysis of neurite length in various groups. As depicted, all test compounds were able to induce neurite outgrowth: PCI24781 (0.1 µM) had the greatest effect with 37% increase compared with that of the 0.1% DMSO. It was followed by JNJ26481585 (0.001 µM) with 34% increase. Trichostatin A was used as a positive control. Quantitative analysis of neurite length was done using ImageJ software. Two fields of view were analyzed per well. Two biological replicates were used per treatment and two independent experiments were done. Error bars represent SEM; *** *p* < 0.001, **** *p* < 0.0001.

**Figure 6 ijms-20-01109-f006:**
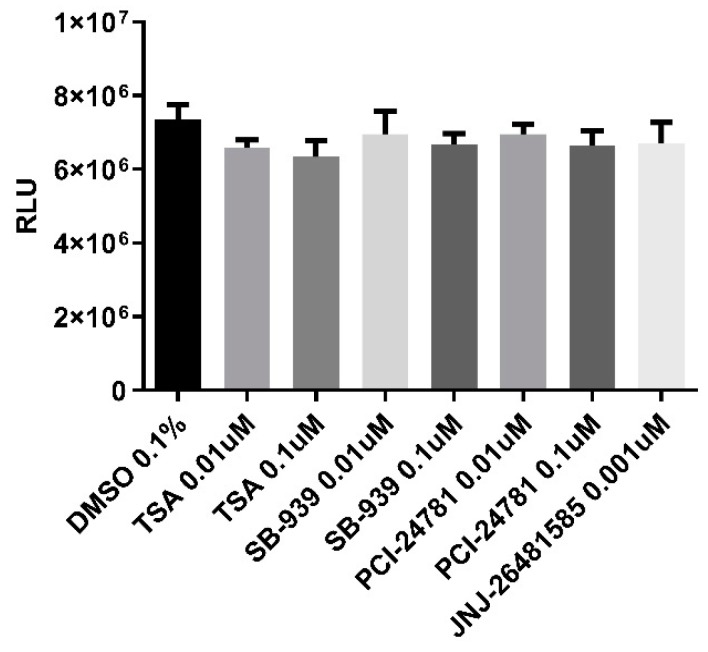
CellTiter-Glo assay showing the effect of test compounds on human neural progenitor cells viability after 24 h of exposure. There was no significant toxicity upon treatments. Data are mean of the experiment results performed in quadruplicate. Error bars represent SD.

**Table 1 ijms-20-01109-t001:** Common compounds activating *BDNF* mRNA expression in both human neural progenitor cells and fibroblasts.

	Compound	Category
*1*	CAY10603	HDAC Inhibitor
*2*	Givinostat (ITF2357)	
*3*	JNJ-26481585 (Quisinostat)	
*4*	PXD101 (Belinostat)	
*5*	M-344	
*6*	PCI-24781 (Abexinostat)	
*7*	LMK 235	
*8*	SB-939 (Pracinostat)	
*9*	Tubastatin A	
*10*	(*S*)-HDAC-42 (AR42)	
*11*	CAY10398	
*12*	Phenylbutyrate·Na	
*13*	SAHA	
*14*	BML-210	
*15*	Valproic acid	
*16*	Apicidin	
*17*	CUDC-907	
*18*	Oxamflatin	
*19*	Trichostatin A	
*20*	MM-102	Methyltransferase Inhibitor
*21*	5-Aza-2′-deoxycytidine (Decitabine)	
*22*	Nicotinamide (Niacinamide)	PARP-1 Inhibitor
*23*	BYK 204165	
*24*	CPTH2	HAT Inhibitor
*25*	2,4-Pyridinedicarboxylic Acid (2,4-PDCA)	Histone Demethylase Inhibitor
*26*	Resveratrol	Natural Compound

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
