# Peer review of "HDAC Inhibitors Induce BDNF Expression and Promote Neurite Outgrowth in Human Neural Progenitor Cells-Derived Neurons"

_ijms, 2019, doi:10.3390/ijms20051109_

Round 1
Reviewer 1 Report
The study offers interesting aspects for understanding the regulation of BDNF expression. A limit of the work is the use of a home made library and sometimes the rationale of the chosen compounds is not fully clear. On the whole the paper is well written. The results are displayed clearly and supported by experimental data. Definitely the determination of HDAC inhibitors on BDNF gene is an interesting result with potential therapeutic application.
Minor points: the authors insert at the end of the text a table of abbreviations but in the text sometimes abbreviations are reported other not. The authors must homogenize the text and, by the way, in the abstract the used acronym must be specified for clarity.
Pag 1 line 42 add references at the end of sentence.
The term disorders is repeated; change brain disorders with brain pathologies
Pag 2 line 62 “…following reasons: Firstly, unlike neuronal loss, synaptic dysfunction and loss are reversible. Secondly, regardless….”
Use the for firstly; change dot with semicolon.
Author Response
1- The authors insert at the end of the text a table of abbreviations but in the text sometimes abbreviations are reported other not.
***Correction is made.
2- The authors must homogenize the text and, by the way, in the abstract the used acronym must be specified for clarity.
***Correction is made.
3- Page 1 line 42 add references at the end of sentence.
***Correction is made.
4- The term disorders is repeated; change brain disorders with brain pathologies
***Correction is made.
5- Page 2 line 62 “…following reasons: Firstly, unlike neuronal loss, synaptic dysfunction and loss are reversible. Secondly, regardless….” Use the for firstly; change dot with semicolon.
***Correction is made.
Point-by-point list of corrections:
1- Line 26 The acronym is updated and described in full.
2- Line 42 The word “disorders” is changed with “pathologies” to reduce the repetitive use of the word “disorder”
3- Line 99 Sentence is removed to reduce redundancy.
4- Line 134 New references are added to refer to the papers which have showed the efficacy of TSA in BDNF enhancement.
5- Line 178 New reference is added to refer to the paper which has demonstrated the effect of TSA on neurite outgrowth.
6- Line 278 A sentence is added to describe the need for further investigation to check whether the demonstrated effect of HDAC inhibitors on neurite outgrowth is directly due to BDNF increase.
7- Line 311 The unnecessary explanation of the acronym “HNPCs” is removed.
8- Line 374 The unnecessary description is removed.
9- Line 380 The unnecessary sentence is removed.
10- Line 413 The acronym is updated.
11- Line 419 The redundant acronym is removed.
12- Throughout the manuscript the “HDACIs” is changed to “HDACis”.
13- Throughout the manuscript the numbering of the references has changed due to the addition of new references.
Reviewer 2 Report
Brain-derived neurotrophic factor (BDNF) has a critical role in neuronal aspects including neurogenesis, synaptic function and plasticity. Therefore, in this study, they performed a screening with an in-house epigenetic library using cultured human neural progenitor cells (HNPCs) and WS1 human skin fibroblast cells, and identified the small compounds capable of increasing the BDNF expression.
They identified HDAC inhibitors including SB‐939, PCI‐24781 and JNJ‐26481585 with high impact on BDNF expression. These selected compounds were also effective in neurite outgrowth in HNPCs-derived neurons.
This study is well-designed. My only concern is that the neurite outgrowth after addition of HDAC inhibitors identified is truly via BDNF production. They should confirm inhibitory effect of Trk (receptor for BDNF) inhibitor when HDAC inhibitors are applied.
Author Response
We thank the reviewer for this valuable suggestion and agree that it would be interesting to carry out the suggested experiment. We have added the following sentence to the Discussion. “Neurite outgrowth might be the direct result of BDNF production; however, this hypothesis cannot be proved unless TrkB signaling pathway is separately blocked when the identified HDAC inhibitors are applied.”
Round 2
Reviewer 2 Report
This version has been improved